# Venous Thrombus Embolism in Polytrauma: Special Attention to Patients with Traumatic Brain Injury

**DOI:** 10.3390/jcm12051716

**Published:** 2023-02-21

**Authors:** Deng Chen, Jialiu Luo, Cong Zhang, Liangsheng Tang, Hai Deng, Teding Chang, Huaqiang Xu, Miaobo He, Dongli Wan, Feiyu Zhang, Mengfan Wu, Min Qian, Wen Zhou, Gang Yin, Wenguo Wang, Liming Dong, Zhaohui Tang

**Affiliations:** 1Department of Trauma Surgery, Tongji Trauma Center, Tongji Hospital, Tongji Medical College, Huazhong University of Science and Technology, Wuhan 430030, China; 2Intensive Care Unit, Trauma Center, Suizhou Central Hospital, Hubei University of Medicine, Suizhou 441300, China; 3Department of Trauma Surgery, Trauma Center, Tianmen First People’s Hospital, Hubei University of Science and Technology, Tianmen 417300, China; 4Department of Emergency Medicine, The People’s Hospital of China Three Gorges University, China Three Gorges University, Yichang 443000, China

**Keywords:** polytrauma, traumatic brain injury, venous thrombus embolism, deep venous thrombosis, pulmonary embolism

## Abstract

Venous thrombus embolism (VTE) is common after polytrauma, both of which are considered significant contributors to poor outcomes and mortality. Traumatic brain injury (TBI) is recognized as an independent risk factor for VTE and one of the most common components of polytraumatic injuries. Few studies have assessed the impact of TBI on the development of VTE in polytrauma patients. This study sought to determine whether TBI further increases the risk for VTE in polytrauma patients. A retrospective, multi-center trial was performed from May 2020 to December 2021. The occurrence of venous thrombosis and pulmonary embolism from injury to 28 days after injury was observed. Of 847 enrolled patients, 220 (26%) developed DVT. The incidence of DVT was 31.9% (122/383) in patients with polytrauma with TBI (PT + TBI group), 22.0% (54/246) in patients with polytrauma without TBI (PT group), and 20.2% (44/218) in patients with isolated TBI (TBI group). Despite similar Glasgow Coma Scale scores, the incidence of DVT in the PT + TBI group was significantly higher than in the TBI group (31.9% vs. 20.2%, *p* < 0.01). Similarly, despite no difference in Injury Severity Scores between the PT + TBI and PT groups, the DVT rate was significantly higher in the PT + TBI group than in the PT group (31.9% vs. 22.0%, *p* < 0.01). Delayed anticoagulant therapy, delayed mechanical prophylaxis, older age, and higher D-dimer levels were independent predictive risk factors for DVT occurrence in the PT + TBI group. The incidence of PE within the whole population was 6.9% (59/847). Most patients with PE were in the PT + TBI group (64.4%, 38/59), and the PE rate was significantly higher in the PT + TBI group compared to the PT (*p* < 0.01) or TBI (*p* < 0.05) group. In conclusion, this study characterizes polytrauma patients at high risk for VTE occurrence and emphasizes that TBI markedly increases the incidence of DVT and PE in polytrauma patients. Delayed anticoagulant therapy and delayed mechanical prophylaxis were identified as the major risk factors for a higher incidence of VTE in polytrauma patients with TBI.

## 1. Introduction

Polytrauma describes patients with two injuries that are greater than or equal to a score of 3 on the AIS and one or more additional diagnoses (pathologic conditions) [1,2]. Polytrauma is characterized by poor outcomes and high mortality due to fatal damage and intractable complications, especially trauma-induced coagulopathy (TIC). Studies have highlighted TIC as a common direct cause of early post-injury mortality (30–50%) [3]; 89% of deaths were bleeding-related. A venous thromboembolism event (VTE) is a pathological outcome related to a coagulation disorder originating from TIC. A VTE consists, principally, of deep vein thrombosis (DVT) and pulmonary embolism (PE). Previous studies have found the incidence of lower extremity DVT to be as high as 20% in polytrauma patients [4], and DVT is also the primary cause of death in patients who survive their initial traumatic injuries [5]. The occurrence of fatal PE is reportedly uncommon after major trauma, but it accounts for about 11.9% of late death in polytrauma patients [6,7].

Accumulating evidence indicates that more than 50% of polytrauma patients have an associated traumatic brain injury (TBI), which is an important factor influencing long-term survival [7,8]. Of note, TBI was a recognized independent risk factor for the development of VTE [9]. According to our clinical practice experience, polytrauma patients with TBI are at a higher risk of developing VTE than polytrauma patients without TBI because early treatment is more difficult due to complicated and intractable conditions.

Previous studies have focused primarily on VTE (DVT or PE) occurrence in either polytrauma or isolated TBI. To our knowledge, there is limited clinical data on VTE occurrence in polytrauma patients with TBI. To address this issue, we performed a clinical study to assess the occurrence of VTE over time and identify risk factors for VTE in polytrauma patients with TBI. The aim of the present study was to test the hypothesis that TBI increases the risk of VTE in polytrauma patients because of the dilemma of early therapeutic interventions.

## 2. Materials and Methods

### 2.1. Patient Selection and Diagnostic Modalities

This was an institutional review board-approved, retrospective, multi-center, observational trial with informed consent. The occurrence of venous thrombosis and pulmonary embolism from injury to 28 days after injury was observed. All patients involved in this study were from the traumatic intensive care unit (TICU) or intensive care unit (ICU) of the Advanced Trauma Center (Level I, verified by the China Trauma Rescue & Treatment Association (CTRTA)) at the Tongji Hospital (Wuhan), Suizhou Central Hospital (Suizhou), and Tianmen People’s Hospital (Tianmen) from May 2020 to October 2021. Inclusion criteria for this retrospective analysis were as follows: (1) age over 18 years; (2) isolated traumatic brain injury or injuries with an AIS score ≥ 3 in ≥ 2 body regions (2AIS ≥ 3) [1]. Patients younger than 18 years, incarcerated, pregnant, and/or mentally ill and patients with a history of VTE or who were treated with delayed presentation due to transfer were excluded from our study. There were 862 patients enrolled in our study, and 15 patients were excluded based on exclusion criteria; eventually, a total of 847 consecutive patients met the eligibility requirements. All enrolled patients were classified into either the polytrauma with TBI (PT + TBI), polytrauma without TBI (PT), or isolated TBI (TBI) group according to their diagnosis. VTE was diagnosed based on the presence of DVT or PE. DVT was defined as an abnormality seen on venous Doppler ultrasound (VDU), such as the presence of dilated, noncompressible veins or intraluminal shadows consistent with thrombosis [10]. PE was defined as a filling defect detected on computed tomography pulmonary angiography (CTPA), which was ordered only on the basis of clinical suspicions, such as the onset of abrupt and unexplained hypoxia, hypotension, tachycardia, or any combination of these [11]. The Pulmonary Embolism Severity Index (PESI) was applied to predict the risk-stratifications of PE. CTPA was recommended for patients with a revised Geneva score ≥ 3 coupled with age-adjusted D-dimer cut-offs, according to guidelines [10]. TBI was defined by the presence of subdural hemorrhage, epidural hemorrhage, focal contusion, or diffuse axonal injury on computed tomography (CT), according to recommended guidelines [12]. Polytrauma was defined as injuries with an AIS score ≥ 3 in ≥ 2 body regions (2AIS ≥ 3), according to the new ‘Berlin definition’ [1]. All enrolled subjects received standardized treatment and management per established TBI and polytrauma guidelines [12,13]. All enrolled subjects received appropriate venous thromboprophilaxis treatment and management based on the latest guidelines [10]. In detail, on the premise of avoiding further injury and progressive cerebral hemorrhage, patients received venous thromboprophilaxis as early as possible. Venous thromboprophilaxis includes mechanical prophylaxis (continuous passive motion, once a day) and chemical prophylaxis (heparinum natricum minor molecularis or nadroparin calcium, once a day).

The present study was approved by the ethics committee at Tongji Hospital, Suizhou Central Hospital, and Tianmen People’s Hospital. Patient consent for data collection was obtained from each patient or the patient’s legally authorized representative.

### 2.2. Data Collected

The patient characteristics data collected were the following: sex; age; mechanisms of injury; pre-hospital comorbidities, including smoking status, hyperlipemia, hypertension, and diabetes mellitus; laboratory values, including activated partial thromboplastin time, D-dimer, and platelets; and post-hospitalization interventions, such as the occurrence of DVT/PE, ICU length of stay (LOS), transfusions, operations, and duration of ventilator use. The patient demographics and characteristics (age, gender, physical condition, GCS, ISS, location(s) of injury, etc.) were assessed upon admission. Laboratory values were assessed within 24 h after injury and post-hospitalization interventions, complications, and mortality were assessed from the time of injury to 28 days after injury. The injury profile included the ISS and AIS for body regions. If consent was obtained, serial VDUs and blood samples for coagulation analysis were obtained. Repeat VDU and blood samples were obtained weekly for the patients in our study. Furthermore, a VDU of both lower extremities was performed from the ankle to the inguinal ligament to evaluate the deep venous system and determine the presence of a DVT.

### 2.3. Study End Points

The primary outcome of this study was the incidence of DVT or PE after polytrauma. Secondary outcomes were the risk factors for DVT in the PT + TBI population and the clinical features.

### 2.4. Statistical Analysis

Data were analyzed with SPSS 22.0 (SPSS Inc., Chicago, IL, USA). Prior to analysis, all data were examined for normality and homogeneity of variance. Descriptive statistics were performed for all variables. Categorical variables were expressed as percentages, and continuous variables were expressed as mean and standard deviation or median and range, as appropriate. Student’s *t*-test and the Mann–Whitney U test were used to compare continuous variables, and χ2 was used to compare categorical variables. Differences between groups on continuous data were analyzed using the one-way analysis of variance (ANOVA) followed by the Bonferroni test, or Kruskal–Wallis test for non-parametric data. Group differences in categorical variables were compared using Pearson’s χ2 or Fisher’s exact test for non-parametric data. A multivariable logistic regression analysis was performed to identify risk factors for DVT in the PT + TBI group. All patients in the PT + TBI group who met the diagnostic criteria for DVT were allocated to the DVT^+^ group. The factors differentiating the two groups were first identified through univariate analysis. Age and covariates with statistical significance (*p* < 0.05) were included in the multivariable logistic regression model. The overall fit of the final model was evaluated using the Hosmer–Lemeshow goodness-of-fit index and the area under the receiver operating characteristic (ROC) curve. The final model expressed odds ratios and 95% confidence intervals. For all tests, *p* < 0.05 was considered statistically significant.

## 3. Results

### 3.1. Patient Demographics and Characteristics

The study cohort consisted of 847 trauma patients who met the inclusion criteria. Among these, 383 patients were diagnosed with polytrauma and TBI (PT + TBI group), 246 with polytrauma without TBI (PT group), and 218 with isolated TBI (TBI group). The demographics and characteristics of the study population are summarized in Table 1. No significant differences were observed between the three groups with respect to basic demographics, including age, sex, and comorbid medical conditions. Particularly, the mean Glasgow Coma Scale (GCS) scores upon presentation were comparable for the PT + TBI and TBI cohorts, and the ISS was similar among the PT + TBI and PT groups (Table 1).

### 3.2. Incidence of DVT

The overall incidence of detectable DVT among all enrolled trauma patients was 26% (220/847). The DVT rate was 31.9% (122/383) in the PT + TBI group, 22.0% (54/246) in the PT group, and 20.2% (44/218) in the TBI group. The difference in DVT rates between the PT + TBI and TBI groups was significant (31.9% vs. 20.2%, *p* < 0.01, Table 1) despite the two groups having similar GCS scores. Furthermore, the DVT rate was significantly higher in the PT + TBI group compared to the PT group (31.9% vs. 22.0%, *p* < 0.01, Table 1) despite a similar ISS. The mean number of days after admission for the first-time DVT diagnosis were 8.8 ± 3.8, 14.6 ± 4.2, and 13.9 ± 4.3 days in the PT + TBI, PT, and TBI groups, respectively. The results indicate that DVT in the PT + TBI group occurred at an earlier time point after admission than in the other two groups (*p* < 0.01, Table 1).

### 3.3. DVT Risk Factors in the PT + TBI Group

Overall, 31.9% of patients in the PT + TBI group (122/383, DVT^+^ subset) developed a DVT during their hospitalization. Given the high incidence of DVT in the PT + TBI population, the potential risk factors for DVT were assessed. Using univariate and multivariate analyses, we examined clinical parameters and defined potential risk factors in all 383 patients in the PT + TBI group, which proved to be significant in predicting an increase in DVT.

In univariate analyses, older age, higher D-dimer levels, higher ISSs, higher GCS scores, and longer duration of ventilator use were associated with a higher risk of VTE in the PT + TBI group. In particular, patients with delayed mechanical prophylaxis and delayed anticoagulant therapy were more susceptible to developing a DVT in the PT + TBI group. In contrast, femoral fracture, skeletal traction, transfusions, the occurrence of complications, arteriovenous catheterization, and operations were not associated with a higher DVT rate. Significant predictive factors for DVT are listed in Table 2.

After univariate analysis, those variables with a *p*-value < 0.05 were selected for multivariate analysis using a multiple logistic regression model. Delayed anticoagulant therapy, delayed mechanical prophylaxis, old age, and higher D-dimer levels were the significant and independent predictive risk factors for DVT occurrence (Table 3). Because no differences were found in analyses of age, level of D-dimer, and longer duration of ventilator use among the PT + TBI, PT, and TBI groups (Table 1), we hypothesized that delayed mechanical prophylaxis and delayed anticoagulant therapy were the major risk factors for the significantly higher DVT rate in PT + TBI patients.

### 3.4. Demographics and Characteristics of Patients Diagnosed with PE

In the current study, the diagnosis of PE was largely based on CTPA, which is the gold standard diagnostic technique for evaluating patients with suspected PE [10]. A total of 96 patients with clinically suspected PE received a CTPA examination, and PE was diagnosed in 59 patients. The incidence of PE in the study cohort was 6.8% (59/847). Of the 59 patients with PE, 38 were in the PT + TBI group (9.9%, 38/383), 11 in the TBI group (5.0%, 11/218), and 10 in the PT group (4.1%, 10/246). Most patients with PE were in the PT + TBI group (64.4%, 38/59), and the PE rate was significantly higher in the PT + TBI group compared to the PT (*p* < 0.01) or TBI (*p* < 0.05) group.

In this study, all patients diagnosed with PE were evaluated with the Pulmonary Embolism Severity Index (PESI) [14]. Overall, 8 patients were categorized as class Ⅰ risk strata (very low mortality risk); 9 patients were categorized as class Ⅱ risk strata (low mortality risk); 17 patients were categorized as class Ⅲ risk strata (moderate mortality risk); 12 patients were categorized as class Ⅳ risk strata (high mortality risk); and 13 patients were categorized as class Ⅴ risk strata (very high mortality risk). Patients with PE categorized as class Ⅲ–Ⅴ risk strata had significantly higher PE severity and mortality risk. As shown in Table 4 and Figure 1, the proportion of patients with a severe PE risk score (class Ⅲ–Ⅴ risk strata) was significantly higher in the PT + TBI group than in other groups (*p* < 0.05).

A total of 31 patients in the PT + TBI group died during their hospitalization, 8 (25.8%, 8/31) of whom were diagnosed with PE. However, only 3 of these deaths could be directly attributed to PE; the remaining deaths were caused by sepsis or multisystem organ dysfunction syndrome. We summarized the clinical data of the three patients who died from PE and found that all deaths occurred in male patients and patients with a severe PESI risk score (class Ⅲ–Ⅴ risk strata). One of the patients who died was placed on a prophylactic inferior vena cava filter (IVCF) before the diagnosis of PE using CTPA. During the early stage (≤3 days) after the initial injury, none of the three patients who died of PE received chemical anticoagulation, and only one received mechanical prophylaxis.

## 4. Discussion

The overall incidence of VTE in our study population and the VTE incidence in our TBI group are in line with previously published results from severely injured trauma patients [4,5,9]. However, few studies have reported the incidence of VTE in polytrauma patients with TBI. To our knowledge, this is the first multi-center trial to investigate the incidence of VTE simultaneously in three groups of trauma patients: PT with TBI, PT, and isolated TBI. Importantly, we verified our hypothesis that the presence of TBI further increases the risk of VTE in polytrauma patients. Moreover, we found that VTE occurred at an earlier time point after admission in the PT + TBI group than in the PT and isolated TBI groups.

In our current study, severe traumatic injuries invariably increased the risk of VTE, which is consistent with previous studies. It has been widely acknowledged that acute trauma increases the rate of VTE and is responsible for approximately 8–12% of VTE episodes in the community [5,15]. Notably, this is the first study to suggest that TBI further increases the risk for VTE in polytrauma patients. One likely mechanism is that, compared to polytrauma patients, TBI increases prolonged immobility in polytrauma patients due to unconsciousness or impairment of brain motor function. Prolonged immobility is a common risk factor for VTE because it leads to reduced blood flow and venous stasis. Venous stasis, along with hypercoagulability and endothelial injury, is also involved in and contributes to the pathophysiology of venous thrombosis [16]. A systemic meta-analysis of epidemiological studies showed a significant increase in the risk for VTE in patients with reduced mobility, and a validation study for a risk assessment model (RAM) of VTE identified immobility as an important predictor for the occurrence of VTE [17].

In contrast, Valle et al. found a similar VTE rate in PT + TBI compared to TBI patient groups, and concluded that TBI did not further increase the risk for VTE in polytrauma patients [18]. This difference may be due to their relatively small sample size (148 cases, single-center trial) and some differences in experimental design. Furthermore, another possible explanation for the lack of observed difference in the VTE rate between the two subgroups is that all enrolled patients used mechanical prophylaxis. However, in our present study, a considerable number of patients could not be treated with mechanical prophylaxis due to the frequent presence of lower extremity fractures or severe soft tissue damage. To date, there are no further studies investigating whether TBI increases the risk for VTE in polytrauma patients.

Notably, we are the first to propose the idea that DVT in the PT + TBI group occurs at an earlier time point after admission than in the PT and isolated TBI groups. This may result from an increased burden and impact on the coagulation system. Recent investigations from TEG analysis have demonstrated that coagulopathy in critical polytrauma patients typically resolved within 24 h, followed by a transition into a hypercoagulable or prothrombotic state [19]. TBI usually leads to serious shock, acidosis, and hypothermia, which further impairs the abnormal coagulation system of polytrauma patients. Moreover, previous studies reported a unique TBI-associated coagulopathy associated with the systemic release of tissue factor and brain phospholipids during blood–brain barrier breakdown [20]. Thus, the coexistence of polytrauma and TBI may accelerate the formation of the prothrombotic state, leading to earlier thrombosis in the PT + TBI group. 

D-dimer is a biomarker of fibrin formation and degradation. In practice, the sensitivity of D-dimer for DVT and PE far exceeds the specificity of the test, especially in older patients. In our study, D-dimer levels were routinely measured upon admission, and the diagnosis of twelve VTE cases was established because of a high D-dimer level; eventually, D-dimer levels ≥5.0 µg/mL during hospitalization were identified as an independent predictor of VTE. A retrospective bioinformatics analysis conducted by Michael et al. identified neurosurgical inpatients who underwent a protocol assessing serum D-dimer levels and had a VDU study evaluating the presence of VTE, and reached the conclusion that the D-dimer protocol was efficient in screening for VTE during hospitalization [21].

We hypothesize that delayed anticoagulant therapy and delayed physical prevention were the major risk factors for DVT in PT + TBI patients since no differences were found among the three groups in terms of age, D-dimer levels, or longer duration of ventilator use. In the present study, patients treated with delayed anticoagulant therapy were more susceptible to developing DVT (Table 2), which indicates that early pharmacological prophylaxis is extremely important for preventing DVT in PT + TBI patients. Nevertheless, to date, no clear guidelines exist with regard to the appropriate initial time at which to start pharmacological prophylaxis in patients with TBI. Many studies have examined the viability of this practice; for example, in neurosurgical patients, antithrombotic prophylaxis with certoparin was determined to be safe and effective, and after initiating early pharmacological prophylaxis, the rate of VTE in TBI patients decreased by approximately 50% [22]. Christopher et al. also found that early pharmacological prophylaxis was useful in controlling the number of complications from VTE and pulmonary embolism [23]. Taken together, it seems reasonable that VTE chemical prophylaxis should be undergone as early as possible. However, due to the lack of reliable recommendations, the initiation of chemical prophylaxis is currently mainly based on stable bleeding conditions in repeated head CT scans or on the physicians’ clinical experience.

Mechanical prophylaxis is a critical approach to VTE prevention, especially for those patients with possibly life-threatening organ hemorrhages (cerebral, digestive, etc.). External mechanical devices, such as graded compression devices or intermittent pneumatic compression devices, have been shown to be effective in preventing DVT [24]. In our study population, the frequent presence of lower extremity fractures or severe soft tissue damage of the lower extremities made it difficult to apply external mechanical devices, which may have contributed to a higher DVT incidence in the PT + TBI group. 

The rate of PE was 6.9% (59/847) in our study, which is much higher than other reported rates of 1.0 to 3.7% in polytrauma patients [11,25]; this higher rate may have resulted from the high rate of CTPA examinations performed in our study population. The diagnosis of PE is usually confirmed with CTPA. CTPA examination is not recommended for patients without any clinical hint of PE. However, given the extreme severity of the disease in the PT + TBI group and the potentially fatal consequences of pulmonary embolism, in our current study, we tried to figure out the true incidence of PE. Therefore, we performed CTPA on some patients who presented with symptoms after weighing the risks and benefits. In this way, we can predict PE risks in advance and make emergency plans. Moreover, some patients with no anticoagulant contraindications may also be treated. The early clinical manifestations of PE are extremely nonspecific and in no way indicative of a PE diagnosis; therefore, PE is one of the most frequently missed diagnoses in primary care. Taken together, we suggest that the frequency and rate of CTPA examinations should be appropriately increased to determine the presence of PE in PT + TBI patients to decrease the incidence of adverse clinical events.

Our study has some limitations, the most important of which is the retrospective nature with all of its restrictions on the study design. In our study, the diagnosis of DVT was mostly based on ultrasound examinations during the patient’s hospitalization; therefore, DVT that occurred as clinically insignificant DVT events may have been missed. Moreover, some patients were excluded because of the inability to perform lower limb ultrasounds. We also neglected to collect data on the time to mobilization and active infection or inflammation, which could contribute to the development of VTE. Given the multi-center design of the study, different kinds of anticoagulants were used, which may have influenced the incidence of VTE [26]. Finally, this study included patients that underwent surgery and those who did not, introducing an additional source of potential bias.

In conclusion, this study characterizes polytrauma patients that are at high risk for VTE and emphasizes that TBI further increases the risk for DVT and PE in polytrauma patients. VTE occurred at an earlier time point after admission in the PT + TBI group than in the PT and isolated TBI groups. The subgroups (PT + TBI patients) demand more special attention in screening for DVT and PE during the early stage of trauma. Delayed anticoagulant therapy and delayed mechanical prophylaxis are identified as the major risk factors responsible for a higher incidence of VTE in polytrauma patients with TBI. Further prospective trials are warranted to evaluate the safety of early anticoagulant and mechanical prophylaxis in polytrauma patients with TBI.

## Figures and Tables

**Figure 1 jcm-12-01716-f001:**
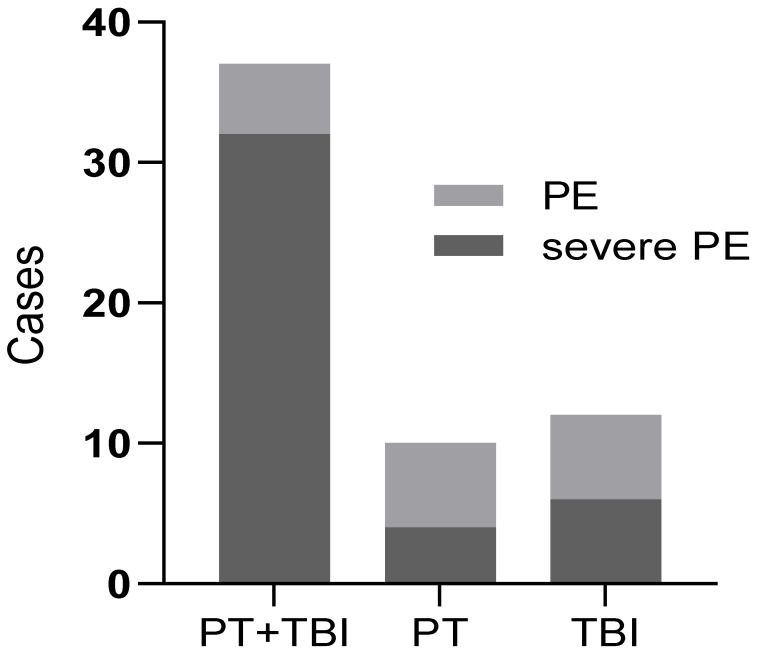
The number of severe pulmonary embolism patients in different groups. PT: polytrauma; TBI: traumatic brain injury; PE: pulmonary embolism.

**Table 1 jcm-12-01716-t001:** Patient demographics and characteristics.

	PT + TBI (n = 383)	PT (n = 246)	TBI (n = 218)	*p*-Value
Mean age (years)	50.3 ± 11.7	49.7 ± 10.4	50.1 ± 11.2	0.806
Males, n (%)	292 (76.2)	178 (72.4)	163 (74.8)	0.550
Hypertension, n (%)	70 (18.3)	41 (16.7)	47 (21.6)	0.404
Diabetes, n (%)	65 (17.0)	38 (15.4)	39 (17.9)	0.773
Hyperlipidemia, n (%)	109 (28.5)	83 (33.7)	59 (27.1)	0.231
Cerebral infarction, n (%)	3 (0.8)	2 (0.8)	3 (1.4)	0.746
Mean ISS	31.2 ± 6.9	30.4 ± 7.5	13.7 ± 3.4	<0.01
Mean GCS	10.4 ± 1.7	-	8.9 ± 2.1	<0.01
Pelvic fracture, n (%)	127 (33.2)	91 (37.0)	-	0.324
Thoracic injury, n (%)	197 (51.4)	134 (54.5)	-	0.457
Abdominal injury, n (%)	177 (44.6)	102 (41.5)	-	0.242
**Transfusion**				
Red cells, n (%)	146 (38.1)	86 (35.0)	70 (32.1)	0.323
Fresh frozen plasma, n (%)	198 (51.7)	133 (54.1)	98 (45.0)	0.126
**Complication**				
Pneumonia, n (%)	118 (30.8)	79 (32.1)	58 (26.6)	0.400
MOF, n (%)	38 (9.9)	30 (12.2)	14 (6.4)	0.108
Sepsis, n (%)	47 (12.3)	37 (15.0)	20 (9.2)	0.158
Ventilation days (days)	8.9 ± 4.6	9.1 ± 4.8	8.2 ± 5.1	0.104
Days in ICU (days)	14.2 ± 4.7	13.8 ± 4.4	13.6 ± 5.6	0.308
Operation, n (%)	292 (76.2)	172 (69.9)	52 (23.9)	<0.01
D-dimer (ug/mL)	10.2 ± 6.0	10.5 ± 5.7	10.1 ± 6.2	0.743
INR	1.18 ± 0.23	1.16 ± 0.27	1.15 ± 0.24	0.313
VTE, n (%)	122 (31.9)	54 (22.0)	44 (20.2)	<0.01
Days of VTE diagnosis (days)	8.8 ± 3.8	14.6 ± 4.2	13.9 ± 4.3	<0.01
Mortality, n (%)	31 (8.1)	10 (4.1)	9 (3.6)	0.049

All results are given as mean ± standard deviation or number (percentage). GCS: Glasgow Coma Scale; ICU: intensive care unit; ISS: Injury Severity Score; MOF: multiple organ failure; INR: international normalized ratio; PT: polytrauma; TBI: traumatic brain injury; VTE: venous thrombotic events. *p* < 0.05 is statistically significant.

**Table 2 jcm-12-01716-t002:** Risk factors for DVT in the PT + TBI group.

Characteristic	DVT(+) (n = 122)	DVT(−)	*p* Value
Age (years)	54.2 ± 13.5	47.3 ± 10.2	<0.01
Males, n (%)	91 (74.6)	201 (77.0)	0.604
Diabetes, n (%)	13 (10.7)	24 (9.2)	0.652
Obesity, n (%)	19 (15.4)	46 (17.6)	0.595
ISS	34.1 ± 7.5	30.1 ± 6.3	<0.01
GCS	9.9 ± 1.5	10.6 ± 1.9	<0.01
**Anatomical location of injury**			
Thoracic injury, n (%)	62 (50.8)	135 (51.7)	0.869
Abdominal injury, n (%)	60 (49.2)	117 (44.8)	0.426
Pelvic fracture, n (%)	38 (31.1)	89 (34.1)	0.567
Cervical/thoracic spinal cord, n (%)	13 (10.7)	35 (13.4)	0.448
Femoral fracture, n (%)	28 (23.0)	53 (20.4)	0.567
**Transfusion**			
Red cells, n (%)	52 (42.6)	94 (36.0)	0.215
Fresh frozen plasma, n (%)	59 (48.4)	139 (53.3)	0.372
**Complication**			
Sepsis, n (%)	19 (16.4)	28 (10.7)	0.119
Pneumonia, n (%)	45 (36.9)	73 (28.0)	0.078
MOF, n (%)	15 (12.3)	23 (8.8)	0.288
INR (%)	1.19 ± 0.22	1.16 ± 0.25	0.257
APTT (s)	40.1 ± 11.6	38.6 ± 10.2	0.200
D-dimer (ug/mL)	12.3 ± 6.2	9.1 ± 5.9	<0.01
Ventilator days (days)	12.4 ± 5.7	6.8 ± 3.4	<0.01
ICU LOS (days)	14.7 ± 5.3	13.9 ± 4.5	0.127
First day of chemical prophylaxis ≤ 3 d, n (%)	19 (15.6)	98 (37.5)	<0.01
First day of mechanical prophylaxis ≤ 3 d, n (%)	51 (41.8)	157 (60.2)	<0.01
Skeletal traction, n (%)	37 (30.3)	67 (25.7)	0.340
Operation, n (%)	97 (79.5)	195 (74.7)	0.304

All values are presented as mean ± standard deviation or number(percentage). APTT: activated partial thromboplastin time DVT: deep vein thrombosis; GCS: Glasgow Coma Scale; ICU: intensive care unit; ISS: Injury Severity Score; LOS: length of stay; MOF: multiple organ failure; INR: international normalized ratio. *p* < 0.05 is statistically significant.

**Table 3 jcm-12-01716-t003:** Logistic regression model for risk factors associated with DVT in PT + TBI patients.

Risk Factors	Univariate HR (95% CI)	*p* Value	Multivariate OR (95% CI)	*p* Value
Age	1.047 (1.016–1.079)	0.003	1.064 (1.018–1.111)	0.006
Mechanical prophylaxis	0.345 (0.144–0.829)	0.017	0.072 (0.014–0.353)	0.001
Chemical prophylaxis	0.299 (0.102–0.874)	0.027	0.173 (0.036–0.825)	0.028
D-dimer	1.108 (1.040–1.180)	0.001	1.111 (1.012–1.220)	0.028
Ventilator days	1.004 (1.001–1.007)	0.006	1.003 (0.999–1.008)	0.137
Femoral fracture	1.286 (0.476–3.470)	0.620	1.794 (0.363–8.853)	0.473
Skeletal traction	1.156 (0.564–4.290)	0.393	0.763 (0.128–4.559)	0.767
ISS	1.067 (1.026–1.109)	0.001	1.046 (0.980–1.117)	0.175
GCS	0.895 (0.813–0.985)	0.024	0.986 (0.837–1.162)	0.864
Diabetes	1.245 (0.337–4.602)	0.743	0.826 (0.117–5.844)	0.848
Obesity	0.783 (0.253–2.424)	0.672	0.724 (0.097–5.382)	0.752

CI: confidence interval; GCS: Glasgow Coma Scale; HR: hazard ratio; ISS: Injury Severity Score; OR: odds ratio. *p*-value < 0.05 is statistically significant.

**Table 4 jcm-12-01716-t004:** Classification of PE severity based on the Pulmonary Embolism Severity Index (PESI).

Risk Strata	PT + TBI (n = 37)	TBI (n = 12)	PT (n = 12)
Class I	2	5	4
Class II	3	4	4
Class Ⅲ	8	2	3
Class IV risk strata	15	1	1
Class V risk strata	9	0	0
Class IV	15	1	1
Class V	9	0	0
Higher PE severity and mortality risk (%)	86.5	25	33.3

PE: pulmonary embolism; PT: polytrauma; TBI: traumatic brain injury.

## Data Availability

All datasets presented in this study are included in the article; further inquiries can be directed to the corresponding author.

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
