# Peer review of "Venous Thrombus Embolism in Polytrauma: Special Attention to Patients with Traumatic Brain Injury"

_jcm, 2023, doi:10.3390/jcm12051716_

Round 1

Reviewer 1 Report

The authors ' Deng Chen et al. manuscript "Venous Thrombus Embolism in Polytrauma: Special Attention to Patients with Traumatic Brain Injury" is devoted to the important issue of polytrauma and brain damage combination and the influence of such a combination on the thromboembolic complications frequency. Authors inspiringly analyzed the results obtained and, among other things, came to interesting conclusions regarding the reasons for the higher DVT  frequency in patients with PT+TBI combination  (delay with mechanical and drug prophylaxis), in addition, the limitations of the study are described very well.

At the same time, there are several questions to the manuscript:

1) The materials and methods indicate the total number of patients 867 - which is probably wrong (847 in results)

2) Why were there no general comparisons of all 3 groups by the Kraskell-Wallis criteria or ANOVA (or chi-square)?

3) Whether the multiplicity  correction was used - this should be indicated in the materials and methods

4) In the section related to DVT risk factors in the PT+TI group, what is meant by univariate analysis - logistic regression?

Reviewer 2 Report

Thanks for this article. This article examines the incidence of VTE in TBI and PT patients in three categories. Hypothesis is a higher incidence on patient with both PT and TBI patients, due to increased influence of PT on coagulation systems and delay of prophylaxis due to TBI.

Authors name is a retrospective, Multicenter study. (see comments on that on methods section).

This article confirms the expected hypothesis and might inform clinicians concerning their decisions on timepoints of beginn of prophylaxis or mobilisation (some limitations here are named later on).

Methods Lane 72: Sentence "All patients... " is not complete. Were screened?

Line 78: Excluded rather than Exlcusions?

Line 81: 867 consecutive patients met the inclusion criteria? This is unlikely. Maybe better present a Figure, explaining how many patients were screened, how many were excluded and why.

Line 85: Did all patient receive Ultrasound? or also only with clinical suspicion? 

Present the results in supplement of the number of Ultrasound examinations and CT for PE. 

In section 2.2. of methods you mention content and therefore Ultrasound. This is confusing? is this a retrospective or a prospective design? 

Results:

For Table 1 consider ANOVA as statistic to compare all three groups. 

How come 32% of patients with TBI only need red blood cells and a significant amount FFPs?

Table 2: What are the two groups? In Table heading both groups are termed DVT?

Section 3.4: Definition of PE is not Necessary in this section. Also how PE was diagnosed should be mentioned in the methods section and not here. Figure 1 does not add much information and can be removed. 

Pulmonary 212 Embolism Severity Index (PESI)  also in Methods

Discussion:

Since the main hypothesis is that delayed prophylaxis and immobilisation is responsible for higher incidence of VTE in the TBI + PT Group maybe the results for dosage and timepoints of initiation could be presented more thoroughly. Moreover, almost 50% of patients did not have mechanical prophylactic measures in the first 3 days. is there an explanation for this? What about the time of mobilisation? 

The Discussion is very long. I think I could significantly be shortened based on the story line. 

Round 2

Reviewer 2 Report

Thanks for your editing on the manuscript and for considering the comments of both reviewers. 

The manuscript has improved. 

Two issues left: 

a) the analysis on ANOVA ist not described in methods. Please add the methodological information. 

b) the table with DVT+ and - : the signs are still very small. Please make the distinction clearer. 
